# Management of Cirrhotic Ascites under the Add-on Administration of Tolvaptan

**DOI:** 10.3390/ijms22115582

**Published:** 2021-05-25

**Authors:** Takuya Adachi, Yasuto Takeuchi, Akinobu Takaki, Atsushi Oyama, Nozomu Wada, Hideki Onishi, Hidenori Shiraha, Hiroyuki Okada

**Affiliations:** Department of Gastroenterology and Hepatology, Graduate School of Medicine, Dentistry and Pharmaceutical Sciences, Okayama University, Okayama 700-8558, Japan; adataku719@yahoo.co.jp (T.A.); yasuto19800125@yahoo.co.jp (Y.T.); at841205@gmail.com (A.O.); nonsan0808@yahoo.co.jp (N.W.); ohnis-h1@cc.okayama-u.ac.jp (H.O.); hshiraha@gmail.com (H.S.); hiro@md.okayama-u.ac.jp (H.O.)

**Keywords:** tolvaptan, liver cirrhosis, ascites

## Abstract

Tolvaptan is a recently available diuretic that blocks arginine vasopressin receptor 2 in the renal collecting duct. Its diuretic mechanism involves selective water reabsorption by affecting the water reabsorption receptor aquaporin 2. Given that liver cirrhosis patients exhibit hyponatremia due to their pseudo-aldosteronism and usage of natriuretic agents, a sodium maintaining agent, such as tolvaptan, is physiologically preferable. However, large scale studies indicating the patients for whom this would be effective and describing management under its use have been insufficient. The appropriate management of cirrhosis patients treated with tolvaptan should be investigated. In the present review, we collected articles investigating the effectiveness of tolvaptan and factors associated with survival and summarized their management reports. Earlier administration of tolvaptan before increasing the doses of natriuretic agents is recommended because this may preserve effective arterial blood volume.

## 1. Introduction

Liver cirrhosis and related complications are still regarded as unresolved issues. Although the prevalence of hepatitis C has decreased with the development of anti-hepatitis C virus (HCV) direct anti-viral agents (DAAs), the incidence of alcohol-related and non-alcoholic steatohepatitis (NASH)-related cirrhosis is increasing [1]. 

The management of ascites and edema is one of the main themes in the treatment of cirrhosis. Diuretics are the first and main agents to control water retention [2]. Spironolactone is the first diuretic used for the management of cirrhotic ascites, followed by furosemide. However, both agents block sodium reabsorption; thus, hyponatremia and a reduction of effective arterial blood volume (EABV) commonly occur as side effects [3]. The arginine vasopressin (AVP) receptor is another target for hydration [4,5]. Terlipressin, an AVP analog with a high affinity for vasopressin-1 (V1) receptors, has been shown to be effective via dilated splanchnic vascular vasoconstriction [4]. Another agent that acts on AVP is tolvaptan, which is a highly selective AVP V2 receptor antagonist [5]. The effectiveness of terlipressin has long been shown, while studies on tolvaptan have been insufficient [2]. 

We would like to focus on the clinical impact of tolvaptan and the management of patients under tolvaptan administration by reviewing recently published articles. 

## 2. Methods

We summarized the mechanisms underlying the development of ascites in cirrhosis, and the management of ascites with diuretics other than tolvaptan, according to textbooks and cirrhosis management guidelines [2,3,6,7,8,9]. We found articles investigating the effect of tolvaptan in cirrhotic ascites via Pubmed, using these search terms: ‘cirrhosis x tolvaptan’, ‘cirrhosis ascites tolvaptan’, and ‘cirrhosis x hyponatremia x tolvaptan’. In Section 4.2.2, we summarized the articles we found and our findings concerning the mechanisms of action, and discussed how to manage cirrhotic ascites under add-on administration of tolvaptan.

## 3. The Mechanisms of Ascites in Cirrhosis

The pathogenesis of Na^+^ and water retention in cirrhosis is related not to an intrinsic abnormality of the kidneys, but rather to extra-renal mechanisms, as kidneys from patients with end-stage liver disease could work appropriately if they are transplanted to patients with normal liver function. Various factors are involved in the appearance of ascites in cirrhosis (Figure 1). Portal hypertension is initiated by increased hepatic resistance to portal blood flow, caused by the distortion of the vascular architecture [10]. Hepatic sinusoidal cellular alterations induce constriction of the sinusoids. Perisinusoidal chronic inflammatory cell infiltration and hepatic stellate cell (HSC) activation induce sinusoidal constriction via cytokines and cell-cell direct contact. Furthermore, vasodilation and vasoconstriction balance are very complex in cirrhosis patients. The splanchnic vascular bed is dilated and hyporesponsive to vasoconstrictors, while several vasoactive molecules are unbalanced. The overflow and underfilling hypothesis is a hypothesized mechanism of ascites formation in cirrhotic patients [11].

In the overflow hypothesis, increased hepatic vascular resistance and sinusoidal pressure induce non-volume-dependent renal Na^+^ retention. Hepatic venous and sinusoidal constriction resulting from liver fibrosis increase hepatic afferent nerve activity. This is followed by the adenosine-mediated hepatorenal reflex. Given that an adenosine A1 receptor antagonist inhibited Na^+^ retention in a cirrhotic rat model, the fibrosis related-reflex is believed to be an important initiator of ascites [12]. This non-volume dependent renal Na^+^ retention can result in total plasma volume expansion and an increase in portosplanchnic bed pressure to overflow ascites. Increased volume of ascites might induce compression of the renal vein, resulting in congestive renal failure [13]. 

In the underfilling hypothesis, increased hepatic vascular resistance and hypoalbuminemia induce transudation of fluid clinically shown to be ascites, resulting in hypovolemia. Given that plasma volume expansion requires effective oncotic pressure, the overflow pathology is usually evident in the early cirrhotic stage in those who maintain serum albumin levels. As cirrhosis progresses, the underfilling pathology would become evident. Factors that cause underfilling of the circulation include various factors other than hepatic vascular resistance and hypoalbuminemia, as cirrhosis is a complex multi-organ disease. Additional factors include: (1) peripheral vasodilation and a blunted vasoconstrictor response to reflex, chemical, and hormonal influences; (2) arteriovenous shunts; (3) impairment of the left ventricular function, called cirrhotic cardiomyopathy; (4) occult gastrointestinal bleeding; (5) volume loss caused by an excessive use of diuretics. The peripheral vasodilation is associated with renal vasoconstriction. Initially, vasodilation occurs in the splanchnic vascular bed; later, it occurs in the systemic circulation, leading to arterial underfilling [14]. 

The vasoconstrictor/antinatriuretic (antidiuretic) systems and the vasodilatory/natriuretic systems are both involved in cirrhotic vascular changes. Representative vasoconstrictors include endothelin, eicosanoids, the renin-angiotensin aldosterone system (RAAS), arginine vasopressin (AVP) (antidiuretic hormone [ADH]), the sympathetic nervous system, and vasodilators (e.g., nitric oxide [NO], glucagon, carbon monoxide [CO], prostacyclin, and endocannabinoids). The vasoconstrictor response occurs as complemental activation due to the underfilling status. The RAAS system is activated from an early stage of cirrhosis via the local wound healing response, and angiotensin II has been shown to be increased in the plasma of cirrhosis patients [15]. RAAS inhibition with angiotensin converting enzyme inhibitors (ACEi) or angiotensin II type 1 receptor blockers (ARBs) has been shown to be effective in the attenuation of liver fibrosis in an experimental model and in clinical trials [16,17]. However, RAAS inhibitors are contraindicated in decompensated cirrhosis because of the risk of hypotension and hepatorenal syndrome [18]. They would be an important treatment option to attenuate the progression of early-stage cirrhosis. NO has been shown to be deeply involved in the pathogenesis of cirrhosis, including hyperdynamic circulation, Na^+^ and water retention, hepatopulmonary syndrome, and cirrhotic cardiomyopathy [19]. Arterial vasodilation leads to a reduction in renal EABV, followed by renal functional deterioration, and hyponatremia. 

## 4. Management of Ascites

The first step of ascites management is nutritional support, especially dietary salt restriction [9,20]. However, if salt restriction is too strict (e.g., <5 g/day), then this may result in hyponatremia and related renal impairment [21]. Moderate salt restriction (5–6.5 g/day) is recommended; however, care should be taken to avoid reducing the daily caloric intake, given that the restriction of salt could be correlated with the reduction of the daily caloric intake [22]. As a next step, diuretics are adopted, as follows. 

### 4.1. Ascites Management with Spironolactone and Furosemide 

The first diuretic agent used for the management of cirrhotic ascites is spironolactone, which is a specific aldosterone antagonist. Given that cirrhotic patients have secondary aldosteronism due to RAAS activation via defected EABV, spironolactone is superior to furosemide in managing cirrhosis [23]. Spironolactone was demonstrated to be more effective (response rate 95%) than furosemide (response rate 52%) in one study [24]. When patients are refractory to spironolactone, furosemide may be added as a next step. However, given that both spironolactone and furosemide are sodium reabsorption restricting agents, severe hyponatremia may occur.

Hyponatremia is a common pathologic status in cirrhosis. Systemic vasodilation and arterial underfilling play a key role in the development of hyponatremia. As explained in the overload theory, sinusoidal obstruction-related sodium and water retention induce volume overload and—as explained in the underfilling theory—RAAS activation induces the same phenomenon. In addition, AVP-induced water retention affects hyponatremia. The majority of cirrhosis patients (90%) show hypervolemic dilutional hyponatremia [25]. The administration of spironolactone and furosemide induces additional hyponatremia, resulting in a more severe state. The severity of hyponatremia has been shown to be correlated with mortality [26]. The addition of spironolactone and furosemide results in a worsening of hyponatremia. 

### 4.2. Ascites Management with AVP Targeting Agents

AVP is a nonapeptide synthesized by the neurosecretory cells in the supraoptic nucleus and the paraventricular nucleus of the hypothalamus [27]. These nuclei have axons that terminate in the neural lobe of the posterior pituitary gland (neurohypophysis) where AVP is released. AVP is released into the systemic circulation in response to extracellular hyperosmolarity, and induces vasoconstriction and antidiuretic action. AVP is also released into the central nervous system (CNS) and acts as neuromodulator that affects many psychiatry functions, such as anxiety, social behavior, learning, and memory [28]. There are three known AVP receptors: V1a, V1b, and V2. These receptors belong to the large rhodopsin-like class-A G-protein-coupled receptor family. V1a receptor is located on the vascular smooth muscle cells to induce vasoconstriction. V1a receptor is also expressed in the CNS to influence a wide variety of brain functions, such as social interaction, anxiety-like behavior, depression, and the circadian rhythm. V1b receptor is expressed in multiple brain regions and peripheral tissues, including the kidney, thymus, heart, lung, and spleen. V2 receptor is located on the basolateral membrane of the distal tubule and collecting ducts in the kidney. The activation of V1a receptors results in vascular constriction, while the activation of V2 receptors results in free water reabsorption in the principal collecting duct cells of the kidneys [29]. After AVP binds to the V2 receptors, intracellular concentration of cyclic AMP (cAMP) increases via adenylyl cyclase activation. Cytoplasmic vesicles carrying the water channel aquaporin 2 (AQP2) are fused to the luminal membrane and increase its permeability to water, resulting in water re-absorption [29]. AVP is increased in cirrhosis, especially in individuals who are hypo-responsive to water load, as assessed by water excretion via urine [30]. Reduced EABV resulting from arterial vasodilation stimulates the secretion of AVP by baroreceptor-mediated non-osmotic stimulation [31]. 

#### 4.2.1. Terlipressin

Terlipressin (*N*^a^-tryglycl-8-lysine-vasopressin) is a synthetic vasopressin analogue that shows vasoconstrictor activity in the splanchnic and systemic vasculature [32]. It is a pro-drug for the endogenous/natural porcine hormone [Lys8]-vasopressin (LVP). Terlipressin has a substitution of lysine for arginine at the eighth position of natural AVP and also has three glycyl residues at the amino terminus [32]. Terlipressin, and especially LVP show high affinity for the V1a receptor (Figure 2). However, terlipressin and LVP also show definite V2 receptor binding affinity. This means that in cirrhosis, terlipressin and its metabolite LVP constrict pathogenic dilated splanchnic vessels, resulting in renal EABV recovery. However, the V2 receptor binding and the activating collecting duct AQP2 pathway induce water reabsorption. 

Terlipressin is now recognized as effective in controlling acute kidney injury by hepatorenal syndrome (HRS-AKI), especially with the concomitant use of albumin infusion, as prescribed in the EASL clinical practice guidelines [2]. Beyond the standard AKI, such as pre-renal, intra-renal, and post-renal, cirrhosis-related renal dysfunction is known as HRS. HRS is defined as renal dysfunction due to reduced EABV, or overactivity of vasoactive agents (e.g., AVP) [33]. HRS-AKI has been newly defined based on the following criteria [34]. 

HRS has been classified into two clinical types: types 1 (HRS-1) and 2 (HRS-2). HRS-1 is defined as the rapid progression of renal dysfunction with a doubling of the initial creatinine to >2.5 mg/dL or a 50% reduction in the initial 24 h creatinine clearance to <20 mL/min in less than 2 weeks. HRS-2 is defined as slowly progressive renal dysfunction in which refractory ascites is the main clinical finding. 

Recently, the International Club of Ascites (ICA) renamed HRS-1 as HRS-AKI and HRS-2 as HRS-NAKI, according to the new definition of AKI in The Kidney Disease: Improving Global Outcomes (KDIGO) guidelines [34]. HRS-AKI is defined by the following criteria: (a) absolute increase in creatinine of ≥0.3 mg/dL within 48 h; and/or (b) urinary output of ≤0.5 mL/kg body weight at ≥6 h; or (c) a ≥ 50% increase in creatinine, using the last available value of outpatient creatinine within 3 months as the baseline value. HRS-AKI is precipitated by bacterial infection, gastrointestinal hemorrhage, large-volume paracentesis without the administration of albumin, or an acute exacerbation of alcoholic liver injury [35,36,37]. 

Several placebo-controlled studies of terlipressin have shown a significant effect on recovery of HRS [38,39]. The effectiveness is definite; however, terlipressin induced several adverse events, including severe respiratory failure. In the newest prospective study, respiratory failure was predominant in the terlipressin group; the incidence was nearly three times that in the placebo group [39]. This effect must be in part associated with the activation of the V2 receptor by terlipressin, followed by increased water reabsorption.

#### 4.2.2. Tolvaptan

Tolvaptan is a highly selective antagonist of the AVP V2 receptor [27]. It was first shown to be effective in body weight reduction and in recovering low sodium levels in patients with hyponatremic chronic heart failure [40]. It was later shown to be effective in controlling the syndrome of inappropriate antidiuretic hormone (SIADH), and autosomal dominant polycystic kidney disease (ADPKD). It has been approved in many countries for the treatment of hyponatremia in heart failure, SIADH, and ADPKD [5]. Although tolvaptan definitely prevented an increase in the size of the ADPKD kidney, a higher discontinuation rate was clearly observed due to adverse events (AEs), including increased transaminase [41]. At doses of 60–120 mg/day in early-stage ADPKD (estimated creatinine clearance ≥ 60 mL/min), a significant increase was observed in the transaminase level in the experimental group compared with the placebo group (4.9% vs. 1.2%) [41]. In later-stage ADPKD (estimated glomerular filtration rate 25–65 mL/min/1.73 m^2^ in 18–55 years of age, 25–44 mL/min/1.73 m^2^ in 56–65 years of age), the administration of tolvaptan was also associated with a higher transaminase level in comparison with the administration of a placebo (5.6% vs. 1.2%) [42]. Based on these reports, US-FDA limits the duration of tolvaptan use to not more than 30 days and to patients with underlying liver disease. However, in Japan, tolvaptan was shown to be effective against cirrhotic ascites without severe AEs at low doses of 7.5 mg/day, and has received national health insurance coverage from 2013 [5]. Accumulating evidence suggests that it can even improve the prognosis of liver cirrhosis, and it is now recognized as an effective agent for the management of cirrhotic ascites [43]. 

##### The Mechanisms of Tolvaptan 

Vaptans are non-peptide vasopressin receptor antagonists, which include V1 receptor antagonists and V2 receptor antagonists [27]. V1 receptor antagonists, such as relcovaptan, have been shown to be effective in controlling Raynaud’s disease, dysmenorrhea, and tocolysis. When first produced in 1992 (when water diuresis in humans was limited), the V2 receptor antagonist mozavaptan was the first vaptan demonstrated to be effective for increasing the serum sodium level [44]. Tolvaptan was shown to be effective for water diuresis in 1998 [45]. In addition to these agents, the use of lixivaptan and satavaptan have also been reported as effective in improving hyponatremia and/or ascites [27,29]. Given that the V2 receptor is located on the principal collecting duct cells of the kidneys, its effects are believed to be specific for renal water reabsorption (Figure 3). Tolvaptan induces electrolyte-free water excretion without changing the total level of electrolyte excretion [46]. In addition to these water-balance-related mechanisms, tolvaptan has been shown to activate antioxidant pathway Nrf2/HO-1 and to restore a damaged renal collecting duct cell line [47]. Given that oxidative stress has been shown to induce renal damage in cirrhosis, the anti-oxidant function might be preferable for the kidneys in cirrhosis patients [48]. 

##### The Clinical Efficacy of Tolvaptan in Cirrhosis

Tolvaptan (7.5-30 mg/day for 7 days) has been shown to have add-on effects to conventional diuretics on ascites in Japanese multicenter randomized control trials (RCTs) (Table 1) [49,50]. Significant body weight loss was also reported in a Chinese multicenter RCT [51]. The effectiveness of tolvaptan has been defined differently in different studies. Recently, a Japanese multicenter study reported that body weight loss of 1.5 kg/week most accurately reflected a reduction of symptoms [52]. 

Given that the effect of tolvaptan has been shown to be around 60%, factors predicting its effect have been reported from several institutions. A prospective multicenter non-interventional, post-marketing surveillance study of 340 Japanese patients showed that body weight reduction was predominant in younger patients with a preserved renal function with low creatinine levels and a high estimated glomerular filtration rate (eGFR) [53]. A preserved renal function, as reflected by the eGFR or creatinine level, was shown to be important in several other reports and should therefore be considered a crucial factor predicting the effect of tolvaptan [54,55,56,57]. While even in chronic kidney disease (CKD) patients (eGFR < 45 mL/min/1.73 m^2^), urinary volume increased and body weight decreased; however, the effect was somewhat lower than in non-CKD patients [58]. A bad renal function is not a contraindication for tolvaptan.

Another later large-scale study with 1098 Japanese patients revealed that a lower serum urea nitrogen (UN) level (<22.4 mg/dL) was the only factor predicting an early tolvaptan response [59]. UN has also been shown to be a predictor of the tolvaptan response in several reports, including a Chinese RCT study with 530 patients [51,53,55,56,60]. Given that a high UN reflects dehydration or decreased renal EABV, maintaining EABV is crucial for drawing the tolvaptan response. UN and/or creatinine have been shown to be important markers in several studies; other markers also have been evaluated.

Low serum sodium levels have been shown to predict a non-response to tolvaptan in several studies [58,61]. The main factor responsible for hyponatremia is the increased production of AVP due to nonosmotic hypersecretion in cirrhosis [29]. In patients with high AVP levels, the standard doses of tolvaptan might be too low to achieve a clinical response. Given that hyponatremia is associated with an increased risk of mortality in cirrhosis [62], the administration of tolvaptan should be considered before severe hyponatremia occurs.

Patients with severe portal hypertension have also been reported to show a low re-sponse to tolvaptan. A high hepatic venous pressure gradient and serum hyaluronic acid levels have been shown to predict a low response to tolvaptan [63].

The spot urine Na/K ratio has been reported as a predictor in several studies [56,64]. Urinary sodium excretion has been shown to be useful for the evaluation of standard na-triuretic agents. Urinary sodium excretion of <78 mmol/day with high-dose diuretic agents resulted in no weight loss in patients with refractory ascites [65]. The urinary Na/K ratio has been shown to be the best candidate marker for defining the daily urinary excretion of sodium [66]. The urinary spot Na/K ratio has been shown to be a marker for predicting the effectiveness of natriuretic agents [67]. A spot urine Na/K ratio of ≥2.5 prior to the administration of tolvaptan has been shown to predict a positive response [64]. Another study showed that a spot urine Na/K ratio of >3.09 indicated a response and that, in combination with a serum urea nitrogen/creatinine ratio of <17.5, predicted a response to tolvaptan in 100% of cases [68].

A greater rate of decrease in urinary osmolality at 4 h after the administration of tolvaptan has been shown to predict a response in several studies [56,58]. Urinary osmolality is maintained with dissolved substances, such as creatinine, urea, urea nitrogen, and sodium. When tolvaptan is effective and the water content in urine increases, urinary osmolality should decrease. This marker is mechanistically sound; however, it does not predict a response prior to the administration of tolvaptan.

In addition to low creatinine levels, a high serum concentration of a vascular tonus-related marker, namely asymmetric dimethylarginine (ADMA), has been shown to be a significant marker in one study [57]. ADMA is a vascular function-related marker reflecting reactive oxygen species (ROS) related vasoconstriction. As shown in Figure 4, ADMA is an endogenous inhibitor of endothelial nitric oxide synthase (NOS) and inhibits acetylcholine-induced vasodilation in endothelial cells [69]. A high serum ADMA has been shown to be correlated with the progression of vascular diseases such as cerebral endothelial damage and cardiovascular diseases [70,71]. ADMA is one of the molecules included in the oxidative stress-related vasoconstriction pathway. The concentration of ADMA levels is regulated by dimethylaminohydrolase (DDAH) activity [71]. DDAH inactivates ADMA via hydrolyzing ADMA to citrulline and dimethylamines. One of the two isoforms of DDAH, DDAH-2 mainly exists in endothelial cells and is extremely sensitive to intracellular ROS [72]. ROS can inhibit DDAH-2 activity, resulting in increased ADMA, which is followed by a reduction of vascular endothelial dilation-related NOS. The ROS-DDAH-ADMA-NOS pathway regulates NO production and endothelial motility. Although high serum ADMA is a marker of an advanced condition of a cerebrovascular disease, it predicts a positive response to tolvaptan. The vasoconstrictive status seems to be included in the tolvaptan-sensitive mechanism. This mechanistic analysis indicates that cirrhosis patients with higher serum ADMA levels might have a vasoconstrictive status with a large EABV and which is sensitive to tolvaptan. Tolvaptan might be effective in non-responsive patients with low ADMA levels when administered in combination with a vasoconstrictor, such as terlipressin.

##### Long-Term Survival after Tolvaptan Administration

Long-term survival after tolvaptan administration has been shown in several reports. A tolvaptan-treated group showed better survival rates than patients who did not receive tolvaptan, although these data were from historical cases [73]. A recent large scale retrospective cohort study using a Japan-wide hospital-based administrative claims database reported that the probability of survival was higher in patients who received tolvaptan [74]. Other diuretics, especially furosemide, have been suggested to be correlated with a poor survival rate along with retardation of the renal function [73]. A positive response to tolvaptan has been shown to predict a patient’s prognosis [68,75,76,77,78]. This is convincing, because cirrhosis patients with hyponatremia or an impaired renal function show a poor prognosis [79], and tolvaptan is able to recover these bad conditions.

Several studies have shown that the administration of tolvaptan had no effect on survival [57,80,81]. These studies showed prognostic factors, such as complications with HCC and low fractional excretion of sodium (FENa) [57], complications with HCC and Child-Pugh class C [80], and a rapid and early decrease in a bioimpedance analysis (defined by intracellular water at 6 h after the first dose of tolvaptan) [81]. These factors are also convincing.

HCC and Child-Pugh class C are both clear factors that predict poor survival [74]. The impact of HCC on survival is affected by the stages of the disease and this might influence the different results of reports on HCC.

FENa is calculated by (urine Na x serum creatinine)/(serum Na x urine creatinine), which reflects the sodium reabsorption status. Although the spot urine mineral concentration is affected by the urine concentration, FENa is relatively constant throughout the day because urinary sodium is adjusted by creatinine excretion [11]. Low FENa is usually a marker of pre-renal low EABV-related AKI because renal Na^+^ and water reabsorption is induced in such situations. A reduced EABV, which indicates low FENa, reflects an advanced cirrhosis-related vascular disorder, which predicts a poor survival probability. A low FENa level has been reported to be a predictor of a poor probability of survival before tolvaptan became available [82]. One means of achieving a better survival rate in patients with low FENa levels might be obtaining a recovery of EABV. Therefore, the administration of terlipressin or the infusion of albumin might be potentially effective approaches to improve EABV. For the management of cirrhosis, continuous human albumin administration has been shown to be effective for achieving long-term survival. The administration of human albumin (40 g twice weekly for 2 weeks, and then 40 g weekly) has been shown to be associated with an improved survival rate in comparison with standard medical treatment [83]. Such an approach would also result in a reduction in UN, which would complement the treatment of tolvaptan in non-responsive patients with high UN values, potentially inducing a response followed by long-term survival.

## 5. Conclusions

Since the approval of tolvaptan as a second-line diuretic for cirrhosis in Japan, many investigations of the response to tolvaptan and its effects on survival have been reported. Given that phenomena related to high UN and low EABV values predict a patient’s response to tolvaptan, and because a positive response is correlated with a good survival rate, the maintenance of appropriate EABV values is critical for cirrhosis patients who receive tolvaptan. The addition of albumin is a recommended approach for maintaining EABV, and terlipressin might be a candidate for combination therapy.

## Figures and Tables

**Figure 1 ijms-22-05582-f001:**
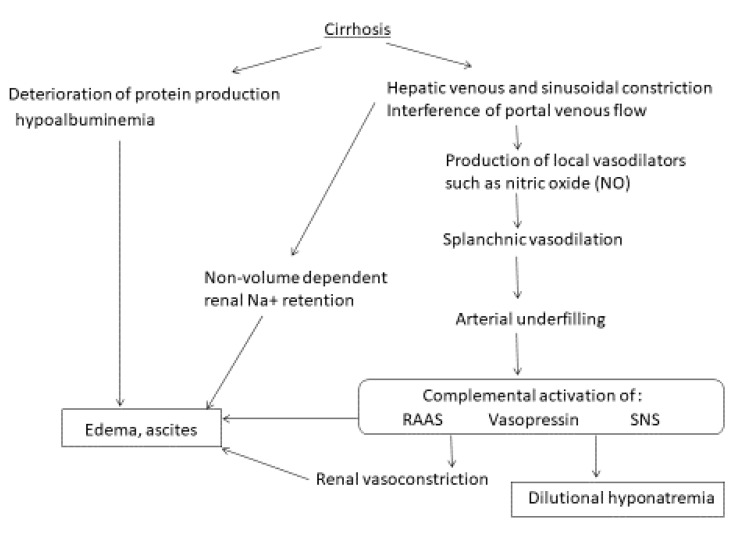
A schematic diagram of the mechanisms of ascites development in cirrhosis. RAAS, renin-angiotensin aldosterone system; SNS, sympathetic nerve system.

**Figure 2 ijms-22-05582-f002:**
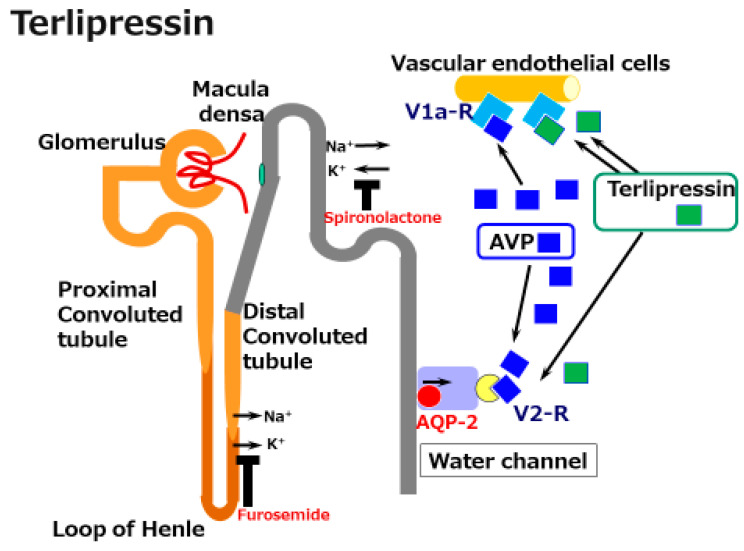
A schematic illustration of the mechanisms underlying the effects of terlipressin in cirrhosis. Splanchnic vascular dilatation is one characteristic of the vascular status in cirrhosis. Terlipressin, a synthetic arginine/vasopressin (AVP) analogue, constricts these dilated vessels and improves effective arterial blood volume (EABV). Although the activation of V2 receptors by vasopressin and terlipressin induces water reabsorption via the water channel aquaporin (AQP)-2 receptor, terlipressin is acknowledged to be a supportive agent for the management of acute kidney injury in cirrhosis.

**Figure 3 ijms-22-05582-f003:**
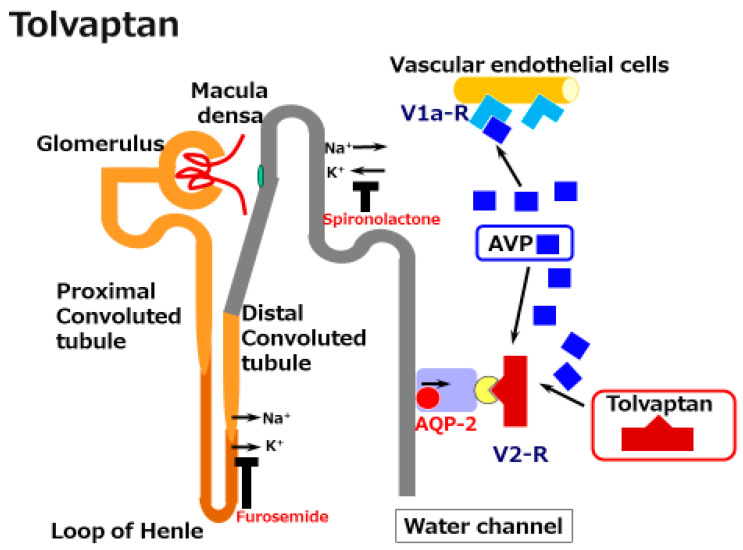
A schematic illustration of the mechanisms underlying the effects of tolvaptan in cirrhosis. Tolvaptan blocks the V2 receptor and reduces the effect of arginine/vasopressin (AVP) in the renal water channel. This does not reduce the preferable effect of AVP on the dilated splanchnic vascular bed.

**Figure 4 ijms-22-05582-f004:**
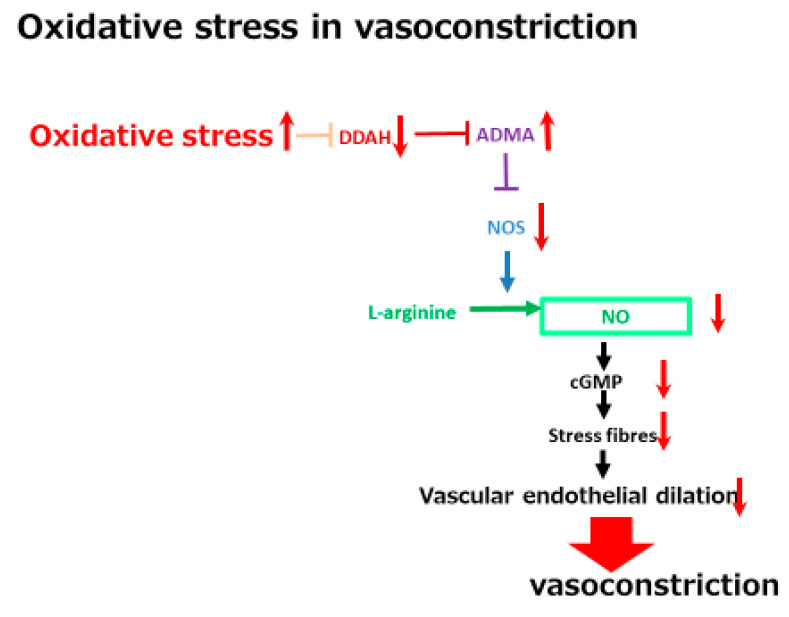
A schematic diagram of oxidative stress in vasoconstriction. The red arrows indicate the changes of the vascular tonus-regulating pathway by oxidative stress. DDAH, dimethylaminohydrolase; ADMA, asymmetric dimethylarginine; NOS, nitric oxide synthase; NO, nitric oxided.

**Table 1 ijms-22-05582-t001:** Summary of placebo-controlled trials for cirrhotic edema with tolvaptan.

Hypothesis	Cohort	Intervention	Outcomes	Reference No.
determine the effects of tolvaptan	164 cirrhosis	Multicenter, double-blind, placebo-controlled. 7-day trial placebo (n = 80), add-on tolvaptan 7.5 mg/day (n = 84)	Tolvaptan decreased the body weight (−1.95 kg in tolvaptan vs. −0.44 kg in placebo)	47
determine the optimal dose of tolvaptan	104 cirrhosis	Multicenter, double-blind, placebo-controlled. 7-day trial placebo (n = 27), add-on tolvaptan 7.5 mg/day (n = 26), 15 mg/day (n = 25), 30 mg/day (n = 26)	Tolvaptan at 7.5 mg/day induced a maximum decrease in body weight with preferable tolerability (−2.31 kg in 7.5 mg/day vs. −1.88 kg in 15 mg/day vs. −1.67 kg in 30 mg/day vs. −0.36 kg/day in placebo)	48
determine the effects of tolvaptan	530 cirrhosis	Multicenter, double-blind, placebo-controlled. 7-day trial placebo (n = 76), add-on tolvaptan 7.5 mg/day (n = 153) 15 mg/day (n = 301)	Tolvaptan decreased the body weight (−2.0 kg in 7.5 mg/day, −2.2 kg in 15 mg/day vs. −1.2 kg in placebo)	49

## Data Availability

Not applicable.

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
