# Peer review of "Management of Cirrhotic Ascites under the Add-on Administration of Tolvaptan"

_ijms, 2021, doi:10.3390/ijms22115582_

Round 1

Reviewer 1 Report

A very interesting review on the usage of Tolvaptan in the management of Cirrhotic Ascites. Well designed and of some interest for the readers of the journal. Well written and with very relevant graphical work (although Figure 1 needs some technical modifications as it is very poor in the technical quality – you can see the pixels in it – but scientifically sound). I have some suggestions still. For example, before acceptance, the authors should provide at least a small methodology section describing how they selected their studies for this review. I see they don t have any COI, but still this is necessary.

Author Response

Response to Reviewer 1

Comments

A very interesting review on the usage of Tolvaptan in the management of Cirrhotic Ascites. Well designed and of some interest for the readers of the journal. Well written and with very relevant graphical work (although Figure 1 needs some technical modifications as it is very poor in the technical quality – you can see the pixels in it – but scientifically sound). I have some suggestions still. For example, before acceptance, the authors should provide at least a small methodology section describing how they selected their studies for this review. I see they don t have any COI, but still this is necessary.

Response:

  1. We re-made Figure 1 to include a higher pixel count than the original article.
  2. We have now added a methods section as “2. Methods”. We found one additional randomized controlled trial (new reference 49) and mentioned it to avoid a COI.

Reviewer 2 Report

This is a well written and interesting review on the management of ascites with different drugs. I have just a few minor suggestions for the authors.

1- Authors review different drugs (not only Tolvaptan) for the management of ascites. This should be reflected in the title of this manuscript.

2- I think the readers would appreciate if the authors include a table with the different placebo-controlled studies cited in the review. The table should content information about the hypothesis, cohort, intervention and outcomes.

Author Response

Response to Reviewer 2

Comments

This is a well written and interesting review on the management of ascites with different drugs. I have just a few minor suggestions for the authors.

  • Authors review different drugs (not only Tolvaptan) for the management of ascites. This should be reflected in the title of this manuscript.

Response:

We changed the title from “Management of cirrhotic ascites under the administration of Tolvaptan” to ”Management of cirrhotic ascites under the add-on administration of Tolvaptan”. Although we reviewed other drugs for controlling cirrhotic ascites, we focused on tolvaptan as an add-on therapy to those other drugs. We would therefore like to include the phrase “under the (add-on) administration of Tolvaptan”. We believe that the term “add-on” suggests that the regimen includes other baseline diuretics.

2- I think the readers would appreciate if the authors include a table with the different placebo-controlled studies cited in the review. The table should content information about the hypothesis, cohort, intervention and outcomes.

We have now added a table showing the placebo-controlled studies of tolvaptan in cirrhosis patients (Table 1).